# Pre-Stroke Antihypertensive Therapy Affects Stroke Severity and 3-Month Outcome of Ischemic MCA-Territory Stroke

**DOI:** 10.3390/diseases12030053

**Published:** 2024-03-03

**Authors:** Lehel-Barna Lakatos, Manuel Bolognese, Mareike Österreich, Laura Weichsel, Martin Müller

**Affiliations:** Department of Neurology and Neurorehabilitation, Lucerne Kantonsspital, 6000 Lucerne, Switzerlandmanuel.bolognese@luks.ch (M.B.); mareike.oesterreich@luks.ch (M.Ö.); laura.weichsel@luks.ch (L.W.)

**Keywords:** stroke, arterial hypertension, dynamic cerebral autoregulation, stroke severity, stroke outcome, antihypertensive therapy

## Abstract

Objectives: Whether different antihypertensive drug classes in high blood pressure (HBP) pre-stroke treatment affect dynamic cerebral autoregulation (dCA), stroke severity, and outcome. Methods: Among 337 consecutive ischemic stroke patients (female 102; median age 71 years [interquartile range, [IQR 60; 78]; NIHSS median 3 [IQR 1; 6]) with assessment of dCA, 183 exhibited the diagnosis of HBP. dCA parameters’ gain and phase were determined by transfer function analysis of spontaneous oscillations of blood pressure and cerebral blood flow velocity. Results: Patients used beta-blockers (*n* = 76), calcium channel blockers (60), diuretics (77), angiotensin-converting enzyme inhibitors (59), or angiotensin-1 receptor blockers (79), mostly in various combinations of two or three drug classes. dCA parameters did not differ between the non-HBP and the different HBP medication groups. Multinomial ordinal logistic regression models revealed that the use of diuretics decreased the likelihood of a less severe stroke (odds ratio 0.691, 95% CI 0.493; 0.972; *p* = 0.01) and that beta-blockers decreased the likelihood of a better modified Rankin score at 3 months (odds ratio 0.981, 95% CI 0.970; 0.992; *p* = 0.009). Other independent factors associated with stroke outcome were penumbra and infarct volume, treatment with mechanical thrombectomy, and the initial National Institute of Health Stroke Scale score. Interpretation: In this cohort of ischemic minor to moderate stroke patients, pre-stroke antihypertensive treatment with diuretics was associated with a more severe neurological deficit on admission and pre-stroke treatment with beta-blockers with a poorer 3-month outcome. The antihypertensive drug class used pre-stroke did not impact dCA.

## 1. Introduction

From a neurological point of view, stroke prevention in patients with high blood pressure (HBP) may appear to be primarily about lowering of the HBP. The class of antihypertensive drug to be used for this purpose is less well-defined, with some guidelines recommending, for example, renin–angiotensin–aldosterone system (RAAS) inhibitors as the first-line agents [1], while others avoid making a clear recommendation for a specific class of drug or make the choice dependent on comorbidities, as long as the main goal of lowering BP is achieved [2,3]. As a result, it remains unclear whether the effects of the different drug classes on the cerebral vasculature and on cerebral autoregulation (CA) contribute significantly to stroke severity and stroke outcome (for an overview, see Llywd et al., 2022) [4]: calcium channel blockers (CCBs), for example, induce cerebral vasodilation and impair CA; some beta-blockers (BBs) may adversely affect CA or cerebral blood flow (CBF); diuretics appear to have no effect on CA or CBF. RAAS inhibitors, such as angiotensin-converting enzyme inhibitors (ACEIs) or angiotensin-1 receptor blockers (ARBs), may shift the CA curve to the left towards lower BP levels, which may make CA more effective at its lower limit, with the result that they may be more protective against ischemia. Overall, however, the literature on pre-stroke antihypertensive medication and stroke severity and outcome is sparse and cannot really contribute to the question of which drug class could be recommended as the first-line therapy in patients with HBP. Pre-stroke use of diuretics [5,6] has been reported to have a beneficial but also a worsening effect on stroke severity and outcome; BBs [7] may not lead to a worse outcome compared to other antihypertensive medications, and ACEIs [8,9] may or may not be associated with a reduction in stroke severity and stroke ischemia volume. CA assessment is a routine procedure in our stroke unit using the bedside method of dynamic cerebral autoregulation (dCA).

Aim of this study: We sought to determine whether different classes of antihypertensive drugs impact dCA differently and whether this corresponds to a reduced stroke severity and better three-month outcomes.

## 2. Methods

This study was conducted adhering to the Declaration of Helsinki, using good standards of clinical practice. This study forms part of a larger trial registered at ClinicalTrials.gov NCT04611672. The corresponding author can provide all data upon request.

The Lucerne Hospital is a large tertiary teaching hospital with a complete stroke center service. All patients with a stroke syndrome receive standardized care, with initially a focused clinical examination followed by a multimodal cranial computed tomography [CT; Siemens Force, Edge or XCeed CT; native], followed by perfusion CT [CTP; postprocessed by Syngo via Rapid AI Software (Version 4.9.1.1) (RAPID)] and CT angiography. If indicated, intravenous thrombolysis and/or arterial thrombectomy follow immediately. All patients diagnosed with stroke syndrome are transferred to the stroke unit for intensive clinical monitoring. The monitoring includes the National Institute of Health Stroke Scale (NIHSS) [10] and modified Rankin score (mRs) [11] assessments upon hospital entry, as well as daily assessments while on the stroke unit and three months after the ischemic event. Blood pressure, heart rate, body temperature, blood glucose levels, and oxygen saturation are closely monitored. In patients receiving medication for hypertension, treatment is usually paused, allowing systolic BP values up to 180–220 mmHg during the first days and not necessarily lowered depending on the patient’s clinical state and the imaging findings. Extensive ultrasound examinations of all brain-supplying arteries, an echocardiogram, and brain magnetic resonance imaging (MRI) with DWI, T2, FLAIR, and SWI sequences on either a Siemens Vida fit (3 Tesla), Siemens Aera (1,5 Tesla), or a Philips Achieva (3 Tesla) follow within 72 h after hospitalization. Brain infarct and penumbra sizes are automatically determined from CTP; infarct size estimation on MRI is calculated by the ABC/2 method, which demonstrated in our hands a good agreement with that by automatic software [12]. Patients whose neurological deficit resolved within 24 h and whose DWI remained negative are classified as having suffered from a TIA. Regardless of whether it had been symptomatic or not, any present cerebral microangiopathy is classified according to the Fazekas scale [13]. The estimation of renal glomerular filtration rate (eGFR) is part of the routine laboratory diagnostics.

### 2.1. Patients

For this study, we prospectively recruited all patients treated at our stroke unit from 1 August 2021 to 31 April 2022 who had a sufficient dCA assessment, irrespective of whether they suffered from HPB or not. Inclusion criteria were as follows: age over 18 years, absence of pregnancy, the presence of a characteristic hemispheric syndrome diagnosed as a supratentorial ischemic stroke in the middle cerebral artery (MCA) territory after initial multimodal imaging and later confirmed by DWI imaging, presence of an ischemic stroke in only just one hemisphere to classify the results into belonging to the stroke-affected hemisphere (AH) and the unaffected hemisphere (UH), transcranial Doppler (TCD) recordings of good quality in the middle cerebral artery at a depth of 45–60 mm, at least at the AH, and determination of dCA within 48 h of the stroke event. Exclusion criteria included the final diagnosis of a stroke mimic, a primary intracranial hemorrhage, as well as a cerebral sinus or vein thrombosis. Regarding the pre-stroke treatment of their HBP, we classified the patients into those having received either diuretics, BBs, CCBs, ACEIs, or ARBs, regardless of whether the patients received additional antihypertensive medication of another class. Thus, a patient receiving a therapy involving two different classes (e.g., a CCB and a diuretic) were allocated in 2 groups (CCBs and diuretics).

### 2.2. dCA Assessment

For extensive details on the dCA assessment, refer to the previous reports [14,15]. We did not stimulate BP or cerebral blood flow velocity (CBFV) by any maneuver. From spontaneous oscillations, we recorded continuously and simultaneously 5 min epochs of BP (Finometer Pro; Finapres Medical Systems, Amsterdam, The Netherlands) and MCA-CBFV (2 MHz probe; MultidopX, DWL; Compumedics, Sipplingen, Germany). Each of these original time series was averaged over one second to generate new time series with less data points. From these new time series, the target variables’ coherence, phase (radian) in, and gain (cm/s/mmHg) were extracted via Transfer Function (TF) estimation using Welch’s averaged periodogram method, with a Hanning window length of 100 s, window overlap of 50%, and total Fast Fourier transformation data length of 300 s. All analysis were performed using Matlab (2023a; Math Works Inc., Natick, MA, USA). The analyzed frequency range was 0.02–0.50 Hz; the results are reported as their respective averages in the very low frequency range (VLF, 0.02–0.07 Hz), low frequency range (LF, 0.07–0.15 Hz), and high frequency range (HF, 0.15–0.5 Hz). In the VLFs and LFs, gain and phase usually exhibit opposite directions. If gain is low, phase is high, and vice versa. An impaired dCA is indicated by a low phase or a high gain, either in the VLF or the LF range.

### 2.3. Statistics

For all data analysis, the Matlab Statistical Toolbox was used. Normally distributed data are reported as mean ± SD and not normally distributed data as median with their interquartile range (IQR, 25th, 75th percentiles). For all group comparisons, the Kruskal–Wallis test was used. The Fisher’s exact test or chi^2^ statistics was used to compare the frequency of categorical variables between groups. Univariate regression analysis and multinomial ordinal logistic regression models were used to identify the outcome-predicting factors. *p* ≤ 0.05 was considered statistically significant.

## 3. Results

During the referred time span, 337 patients (102 female; median age 71years [IQR 60; 78]; NIHSS median 3 [IQR 1; 6]) were recruited. Arterial hypertension was diagnosed in 183 patients. A total of 220 patients had a minor (NIHSS ≤ 5), 82 had a mild to moderate (NIHSS 6–15), and 35 had a major (NIHSS ≥ 16) stroke [16]. The patients without HBP (Table 1 and Table 2) were significantly younger (*p* = 0.0000) and suffered less frequently from diabetes mellitus (*p* = 0.01). The incidence rates of female or male sex, dyslipidemia, current smoking, atrial fibrillation, transient ischemic attack or stroke, lacunar/non-lacunar stroke, and intravenous thrombolysis or/and arterial thrombectomy with actual stent retriever systems did not differ among the non-hypertensive and the different hypertensive patient cohorts. Antihypertensive monotherapy was rare and a combination therapy frequent; BBs were mostly combined with diuretics and/or ARBs, CCBs with diuretics and/or ARBs, diuretics with BBs and ARBs, ACEIs equally with BBs and/or diuretics and/or CCBs, and ARBs with BBs and/or CCBs. On admission, the systolic BP, the penumbra, and the infarct core on CTP were not different between the non-hypertensive and hypertensive groups. The initial NIHSS score and the initial mRs did not significantly differ between the non-hypertensive patients and the hypertensive patient groups. Regarding all other parameters, MRI stroke volume was the lowest (but not significantly) in the group treated with ACEIs, and the mRs at three months was not different between the different patient groups.

Regarding dCA, 201 recordings were performed on day 1 and 136 on day 2. There were no differences in phase or gain between the different groups with or without HBP (Table 3 and Table 4). As indicated by the provided reference values from our healthy control population, phase and gain in the stroke patients were similarly reduced over all subgroups. Comparing phase and gain in the AH of the non-hypertensive patients with all hypertensive patients, the VLF phase was 0.82 [IQR 0.59; 1.02] vs. 0.078 [IQR 0.55; 1.04; *p* = 0.45], LF phase 0.57 [IQR 0.38; 0.80] vs. 0.65 [IQR 0.45; 0.88; *p* = 0.08], HF phase 0.20 [IQR 0.03; 0.38] vs. 0.2 [IQR 0.05 0.38; *p* = 0.78], VLF gain 0.28 [IQR 0.14; 0.43] vs. 0.23 [IQR 0.14; 0.38; *p* = 0.07], LF gain 0.41 [IQR 0.30; 0.62] vs. 0.37 [IQR 0.26; 0.52; *p* = 0.09], and HF gain 0.51 [IQR 0.40; 0.70] vs. 0.47 [IQR 0.36; 0.63, *p* = 0.06]. Overall, phase was similar in the hypertensive and non-hypertensive patients while gain showed a trend to be lower in the hypertensive patients. The results in the UH were similar. NIHSS at entry was 3 [IQR 1; 8] in the non-hypertensive patients and 3 [IQR 1; 7] in the hypertensive ones (*p* = 0.32); mRs at three months was 1 [IQR 0; 1] in the hypertensive and 0 [IQR 0; 2] in the non-hypertensive patients (*p* = 0.17).

### 3.1. Regression Analysis

#### Stroke Severity

For regression analysis, we predefined several variables as independent ones apart from the medications (Table 5, Figure 1). Stroke severity as indicated by the increasing NHISS score was predicted in univariate regression analysis by the use of BBs, use of diuretics, age, the systolic BP at admission, the eGFR, and the presence or absence of extracranial large vessel disease (≥50% stenosis or occlusion of any internal carotid artery), but mostly by the penumbral and stroke core volumes on CTP. In a multinomial ordinal logistic regression analysis model, with the stroke categories minor, mild to moderate, and major stroke (according to the NIHSS) as the dependent variables and age, BBs, diuretics, large vessel disease, penumbra and stroke core volume on CTP, systolic BP on admission, and eGFR as the independent variables, the odds ratios to be allocated in the better stroke category (minor vs. mild to moderate vs. major) were as follows: 0.978 (95% CI 0.966; 0.992; *p* = 0.01) per each increasing year of age, 0.970 (95% CI 0.960; 0.974; *p* = 0.001) per increasing mL penumbra on CTP, 0.973 (95% CI 0.957; 0.981; *p* = 0.002) per increasing mL stroke volume core on CTP, 0.997 (95% CI 0.991; 1.00; *p* = 0.05) per increasing mmHg systolic BP on admission, 1.45 (95% CI 1.01; 2.021; *p* = 0.03) for the presence of extracranial large vessel disease, and 0.691 (95% CI 0.493; 0.972; *p* = 0.01 for the pre-stroke use of diuretics. The other variables were not significant.

Neither any medication nor the CTP findings predicted the dCA parameters’ phase and gain in any of the frequency ranges. Only the systolic BP on admission was predictive for phase in the AH in the VLF range [R^2^ 0.014, beta −0.002 (95% CI −0.004, −0.0002), F-statistic 4.38, *p* = 0.03]. Phase in the VLF and LF range did not predict NIHSS at admission.

### 3.2. Outcome

Outcome prediction variables consisted again of the medication groups and some predefined variables, which have been described as associated with stroke outcome (Table 6). In the univariate regression analysis (Table 6, Figure 2), the outcome as indicated by the mRs at 3 months was predicted by the pre-stroke use of BBs and diuretics, by age, penumbra and infarct volume on CTP, MRI infarct volume, the NIHSS score on admission, the use of iv lysis and/or mechanical thrombectomy, the history of atrial fibrillation, and the eGFR. Including all the significant variables into one multinomial ordinal logistic regression model as the independent variables and the mRs at three months as the dependent variable, the odd ratios to be allocated in the better mRs group were as follows: 0.981 (95% CI 0.970; 0.992; *p* = 0.009) for the use of BBs, 1.005 (95% CI 1.002; 1.008; *p* = 0.01) for each mL penumbra, 0.984 (95% CI 0.978; 0.998; *p* = 0.01) for each mL stoke volume on CTP, 0.982 (95% CI 0.976; 0.988; *p* = 0.008) for each mL MRI stroke volume, 0.542 (95% CI 0.439; 0.671; *p* = 0.002) for the use of mechanical thrombectomy, and 0.778 (95% CI 0.755; 0.801; *p* = 0.0001) per each point on the NIHSS scale. Thus, over the whole mRs range, most independent variables showed an odds ratio to be less likely in the better group or, in other words, to be in the poorer mRs group. A secondary intracerebral hemorrhage within the ischemic infarct occurred in 11 patients (3.3%), 4 in the non-hypertensive group, 7 in the hypertensive one without showing a preference of one medication group.

## 4. Discussion

The aim of this study was to investigate the potential influence of pre-stroke antihypertensive medication on dCA and its impact on acute stroke severity and stroke outcome at the 3 months follow-up. Our main findings are as follows.

In our patients, the antihypertensive pre-stroke medication had no recognizable effects on dCA, as the dCA impairment was not different between the patients without and with HBP. Probably, any effect still present due to the drug’s effectiveness on BP control was overridden by the stroke itself. In healthy persons, antihypertensive treatment can impair dCA [4], as outlined in the introduction section. In treated hypertensive patients, an impaired dCA can normalize if the BP is successfully reduced at an early disease stage but can also remain impaired if HBP is in its chronic stage [14]. However, if an ischemic stroke occurs, dCA is impaired by the stroke itself. Impairment of dCA, as indicated by transfer function analysis, typically occurs within hours after the stroke onset and does not deteriorate significantly further in the next days [17]. One could therefore speculate that the discontinuation of the antihypertensive medication is of benefit to avoid further deterioration. Of greater relevance for dCA impairment is infarct size, with greater impairment in the condition of larger brain infarcts; such dCA impairment is usually present for 5 days before it gradually resolves [17,18]. On average, infarct size was relatively small in all our patient groups, leading to dCA impairments of similar extent across all groups. The only hemodynamic difference was that the cerebrovascular resistance (pulsatility index as a marker of it) in the hypertensive patients was higher than in the non-hypertensive patients, a result to be explained by either the presence of the HBP disease and/or age [14].

In our patients, the antihypertensive pre-stroke medication with diuretics worsened the stroke severity in the ordinal logistic regression model we used: the deterioration effect was equal from the minor to the mild to moderate category and from the mild to moderate category to the major stroke category. Additionally, it should be noted that the use of BBs in the univariate regression analysis also predicted a poorer NIHSS score at admission. Our observations are in agreement with previous studies by Renner et al. [6] and Maïer et al. [19] but in contrast to the study by Tziomalos et al. [5]. Renner et al. [6] found that the use of diuretics was an independent risk factor for a more severe stroke in patients who were more severely ill (median NIHSS 14) than our patients; their suggested hypothesis was that diuretics lead to a hypovolemic state by reducing systemic blood volume. Maïer et al. [19] reported that BBs were without effect on the NIHSS at admission but that the patients using diuretics and CCBs had a more severe NIHSS at admission than non-users of diuretics. Contrary to the findings of Renner et al. [6], Tziomalos et al. [5] reported that the patients with pre-stroke use of diuretics had a median admission NIHSS of 2 while all other antihypertensive drug classes including BBs were without effect on the initial median NIHSS of 12. The use of BBs and/or diuretics may on the other hand hint towards severe cardiac diseases, which are associated with a more severe acute stroke severity [7,20]. Overall, in our cohort of slightly to moderately handicapped acute stroke patients, the effect of BBs and diuretics on stroke severity is small compared to the relevance of penumbra and infarct core size on initial CTP.

In our patients, the multinomial logistic regression analysis indicated that the pre-stroke use of BBs was associated with a poorer outcome, while the use of CCBs, diuretics, ACEIs, and ARBs were not. In the univariate analysis, however, diuretics did affect the outcome negatively. The literature is inconsistent with regard to outcomes. Maïer et al. [19] reported in patients having undergone endovascular therapy that those who used ACEIs or ARBs as pre-stroke treatment had a lower NIHSS at admission compared to those who were not treated with ACEIs or ARBs, but the outcome was equal between these two patient groups (treatment with ACEIs/ARBs vs. without). Other reports [7,8] did not find a difference between the different antihypertensive drugs regarding outcome.

We did not find a convincing positive effect of ACEIs or ARBs on dCA, which is a bit disappointing as both have been shown in animal models to either reduce infarct size or increase CA resistivity against lower BP values [4,21,22,23,24]. Thus, the hypothesis that dCA might be one mechanism to explain drug-specific outcome differences cannot be supported. We, therefore, hypothesize that in our stroke population, the stroke event has overridden any measurable drug effect on dCA.

Two variables exhibited surprising results: first, the effect of penumbra on CTP, which indicated with increasing penumbral tissue a better outcome, and second, the result of mechanical thrombectomy, which indicated that the use of thrombectomy will result in a poorer mRs group. We explain these results as a consequence from our patient selection strategy for thrombolysis. Patients with a larger penumbra are more likely to receive thrombolytic therapy with its overall benefits [25]. Patients who undergo thrombectomy are clinically in a severe neurological state, which will lead to a poor mRs group if the procedure is not performed. Endovascular treatment improves the outcome especially if the modern stent retriever systems are used, but the incomplete recovery from such severe deficits is often accompanied by a still poorer outcome compared to patients with, e.g., an initial minor stroke [25,26]. Thus, we see the thrombectomy results in our analysis still as a marker of stroke severity.

## 5. Study Limitation

The major limitation is that our patients represent, on average, a less severely ill stroke population with low infarct volume. This is the most likely reason why dCA was without outcome relevance in our patients, which is in clear contrast with other studies (summarized in Nogueira et al. [18]), which were performed on patients with on average moderate to major strokes with larger ischemic stroke volumes. A second limitation of our study is its retrospective design. The advantage of our approach is the inclusion of consecutive non-hypertensive patients into the analysis to allow for a comparison with another stroke population to see how good or bad the different BP regimens have been so far. A follow-up study of patients with high blood pressure may be a more drug-focused approach within a HBP population [27] but would miss the comparison to non-hypertensive stroke patients. A population-based register analysis might be the best approach.

## 6. Future Perspective

Hypertension impacts various end-organs differently. It is suggested that distinct medical disciplines, including cardiology, nephrology, endocrinology, and neurology, will have a different look at HBP therapy regarding the most desirable first-line therapy [28,29,30,31]. Hence, further research must focus on determining the specific patient information necessary to construct an end-organ risk profile. This profile will be critical in determining which HBP therapy should be the primary medication.

## 7. Conclusions/Summary

Arterial hypertension is a major stroke risk factor, and its therapy is an essential stroke prevention strategy. Whether one or the other antihypertensive drug class will provide a lower risk of a poorer stroke outcome is under debate. In our cohort of stroke patients, those who received a diuretic or a BB in the pre-stroke HBP management fared worse during the acute stroke condition and exhibited a poorer mRs score three months after the stroke event.

## Figures and Tables

**Figure 1 diseases-12-00053-f001:**
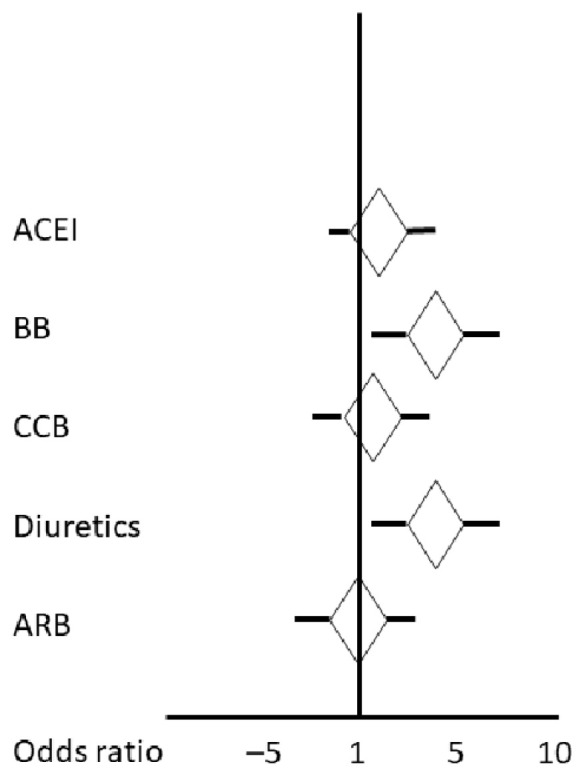
Unadjusted odds ratios of each drug class for worsening stroke severity calculated from the univariate regression analysis. ACEI: angiotensin-converting enzyme inhibitor; ARB: angiotensin-1 receptor blocker; BB: beta-blocker; CCB: calcium channel blocker.

**Figure 2 diseases-12-00053-f002:**
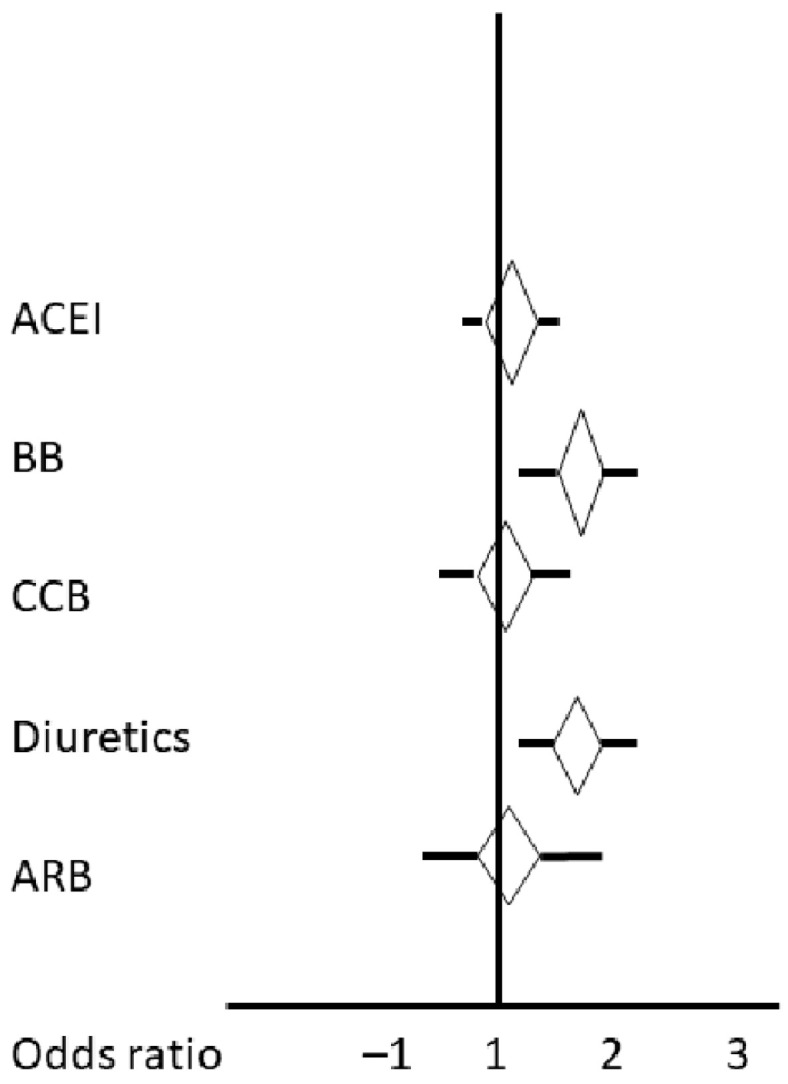
Unadjusted odds ratios of each drug class for worsening modified Rankin scale at 3 months calculated from the univariate regression analysis. ACEI: angiotensin-converting enzyme inhibitor; ARB: angiotensin-1 receptor blocker; BB: beta-blocker; CCB: calcium channel blocker.

**Table 1 diseases-12-00053-t001:** Baseline characteristics of the study population of stroke patients. In total, 154 did not suffer from high blood pressure (BP) and 183 suffered from high blood pressure and were treated with one or more different classes of antihypertensive drugs. General medical aspects.

	No High BP (*n* = 154)	High BP (*n* = 183)				
		Beta-Blockers	Calcium Channel Blockers	Diuretics	RAAS Inhibition	
					ACE Inhibitors	AT-1 Receptor Inhibitors
N	154	76	60	77	59	79
Sex (male/female)	111/43	50/26	41/19	49/28	41/18	54/25
Age (years)	62.5 [IQR 51; 72] *	76 [IQR 67; 82]	75.5 [IQR 71; 83]	76 [IQR 71.5; 82]	72 [IQR 65; 77]	74 [IQR 67; 80]
Diabetes mellitus	23 **	27	19	26	21	24
Dyslipidemia	107	67	57	64	52	68
Smoking	42	15	13	14	19	12
Large vessel disease	27	13	16	17	7	17
Ischemic heart disease	11	38	17	32	20	29
Atrial fibrillation	18	27	13	21	11	15
Left ventricular ejection fraction (%)	60 [IQR 55; 64]	60 [IQR 55; 64]	60 [IQR 55; 64.5]	60 [IQR 56; 65]	60 [IQR 55; 65]	60 [IQR 55; 63]
Estimated glomerular filtration rate (mL/min/1.73 m^2^)	82 [IQR 61; 91]	83 [IQR 67; 90]	82 [IQR 69; 91]	85 [IQR 76; 91]	82 [IQR 71; 91]	80 [IQR 73; 91]
Antihypertensive therapy with 0/1/2/3/>3 different drug classes	0	0/17/29/22/8	0/9/23/20/8	0/6/36/27/8	0/19/29/7/4	0/18/34/23/4

ACE: angiotensin-converting enzyme; AT: angiotensin; Large vessel disease: ≥50% stenosis or occlusion of the carotid artery system; RAAS: renin–angiotensin–aldosterone system; Antihypertensive therapy: 0, no medication; 1, number of patients using the drug class of the respective column only; 2, number of patients using drug class of the respective column plus one other class; 3, number of patients using drug class of the respective column plus two others; >3, the drug class of the respective column plus more than 2 others. * *p* = 0.0000; ** *p* = 0.01.

**Table 2 diseases-12-00053-t002:** Baseline characteristics of the study population of stroke patients without and with high blood pressure (BP). Neurological aspects.

	No High BP (*n* = 154)	High BP (*n* = 183)				
		Beta-Blockers	Calcium Channel Blockers	Diuretics	RAAS Inhibition	
					ACE Inhibition	AT-1 Receptor Inhibitors
*N*	154	76	60	77	59	79
Number of transient ischemic attack/stroke	20/134	11/65	10/50	18/59	6/53	21/58
National Institute of Health Score (NIHSS) at entry	3 [IQR 1; 6.5]	3 [IQR 1; 7.5]	3.5 [IQR 1; 10]	3 [IQR 2; 8]	2 [IQR 1; 3.75]	3 [IQR 1; 5]
Modified ranking score (mRs) at entry	2 [IQR 1; 3.5]	2 [IQR 1; 4]	3 [IQR 2; 4]	3 [IQR 2; 4]	2 [IQR 1; 3]	2 [IQR 1; 3]
CTP infarct core (mL)	0 [IQR 0; 0], range 0–90	0 [IQR 0: 0], range 0–63	0 [IQR 0; 0], range 0–155	0 [IQR 0; 0], range 0–63	0 [IQR 0; 0] range 0–63	0 [IQR 0; 0] range 0–155
CTP penumbra (mL)	0 [IQR 0; 39.2], range 0–365	0 [IQR 0; 0], range 0–221	0 [IQR 0, 0], range 0–372	0 [IQR 0, 0], range 0–221	0 [IQR 0; 9] range 0–187	0 [IQR 0; 26.3] range 0–372
Number of intravenous lysis (total *n* = 123)	56	20	18	27	18	25
Number of mechanical thrombectomy (total *n* = 65)	30	21	10	18	9	14
MRI infarct size (mL)	0.97 [IQR 0.2; 6.97]	0.70 [IQR 0.12; 8.65]	0.77 [IQR 0.01; 7.00]	0.51 [IQR 0.12; 7.90]	0.30 [IQR 0.01; 2.94]	0.56 [IQR 0.05; 5.40]
Lacunar/non-lacunar stroke	22/132	20/56	10/50	9/68	20/39	15/64
mRs 3 month	0 [IQR 0; 2]	1 [IQR 0; 3]	1 [IQR 0; 3]	1 [IQR 0; 3]	1 [IQR 0; 3]	1 [IQR 0; 2]

ACE: angiotensin-converting enzyme; AT: angiotensin; CTP: computed tomography perfusion; MRI: magnetic resonance imaging; RAAS: renin–angiotensin–aldosterone system. Of note, many small infarcts are too small to be recognized by CTP (penumbra and core); the median of such lesions is 0 as well as its IQR; we included the total range to indicate that there were not only small infarctions (unmeasurable for CTP) but also some larger ones.

**Table 3 diseases-12-00053-t003:** Dynamic cerebral autoregulation parameter in the stroke-affected hemisphere.

Variable	Reference Values from [14] in Mean ± SD	No High BP (*n* = 154)	High Blood Pressure (*n* = 183)				
			Beta-Blockers	Calcium Channel Inhibitors	Diuretics	Renin–Angiotensin–Aldosterone System Inhibitors	
						ACE Inhibitors	AT-1 Receptor Inhibitors
Mean velocity (cm/s)		51 [IQR 43; 63]	45 [IQR 39; 51.5]	47.5 [IQR 42; 57]	44.5 [IQR 37.5; 53]	45 [IQR 38.5; 54]	44 [IQR 38; 50]
Pulsatility index		0.86 [IQR 0.75; 0.97] *	1.06 [IQR 0.87; 1.24]	1.05 [IQR 0.93; 1.16]	1.06 [IQR 0.90; 1.23]	1.04 [IQR 0.89; 1.14]	1.00 [IQR 0.85; 1.20]
Coherence							
−VLF	0.47 ± 13	0.59 ± 0.15 0.61 [IQR 0.51; 0.73]	0.58 ± 0.13 0.56 [IQR 0.48; 0.70]	0.56 ± 0.11 0.57 [IQR 0.47; 0.70]	0.56 ± 0.12 0.58 [IQR 0.48; 0.69]	0.58 ± 0.14 0.64 [IQR 0.50; 0.70]	0.59 ± 0.12 0.58 [IQR 0.50; 0.72]
-LF	0.73 ± 0.15	0.67 ± 0.17 0.70 [IQR 0.59; 0.83]	0.63 ± 0.16 0.65 [IQR 0.52; 0.75]	0.60 ± 0.15 0.59 [IQR 0.50; 0.73]	0.60 ± 0.15 0.63 [IQR 0.53; 0.75]	0.64 ± 0.14 0.67 [IQR 0.53; 0.80]	0.65 ± 0.15 0.68 [IQR 0.55; 0.78]
-HF	0.65 ± 0.14	0.63 ± 0.14 0.68 [IQR 0.60; 0.77]	65 ± 0.15 0.65 [IQR 0.55; 0.75]	0.66 ± 0.16 0.61 [IQR 0.51; 0.69]	0.63 ± 0.15 0.64 [IQR 0.54; 0.70]	0.67 ± 0.12 0.67 [IQR 0.54; 0.73]	0.66 ± 0.15 0.69 [IQR 0.58; 0.75]
Gain (cm/s/mmHg)							
-VLF	0.27 ± 0.29	0.31 ± 0.21 0.28 [IQR 0.14; 0.43]	0.31 ± 0.20 0.23 [IQR 0.14; 0.44]	0.25 ± 0.17 0.27 [IQR 0.15; 0.46]	0.26 ± 0.19 0.25 [IQR 0.15; 0.44]	0.30 ± 0.19 0.28 [IQR 0.17; 0.51]	0.24 ± 0.18 0.25 [IQR 0.14; 0.35]
-LF	0.69 ± 0.41	0.54 ± 0.32 0.41 [IQR 0.30; 0.62]	0.47 ± 0.22 0.41 [IQR 0.29; 0.54]	0.39 ± 0.21 0.40 [IQR 0.25; 0.53]	0.46 ± 0.24 0.41 [IQR 0.31; 0.52]	0.42 ± 0.24 0.40 [IQR 0.25; 0.53]	0.40 ± 0.22 0.41 [IQR 0.34; 0.58]
-HF	0.82 ± 0.51	0.56 ± 0.24 0.51 [IQR 0.40; 0.70]	0.59 ± 0.30 0.53 [IQR 0.39; 0.67]	0.52 ± 0.29 0.44 [IQR 0.36; 0.63]	0.54 ± 0.33 0.52 [IQR 0.35; 0.64]	0.55 ± 0.26 0.48 [IQR 0.39; 0.63]	0.52 ± 0.26 0.48 [IQR 0.38; 0.62]
Phase (radian)							
-VLF	1.12 ± 0.35	0.81 ± 0.36 0.82 [IQR 0.59; 1.02]	0.83 ± 0.39 0.79 [IQR 0.60; 0.96]	0.82 ± 0.40 0.81 [IQR 0.64; 0.95]	0.82 ± 0.35 0.69 [IQR 0.49; 1.04]	0.80 ± 0.43 0.70 [IQR 0.53; 0.93]	0.82 ± 0.35 0.81 [IQR 0.55; 1.07]
-LF	0.74 ± 0.21	0.63 ± 0.31 0.57 [IQR 0.38; 0.80]	0.64 ± 0.36 0.62 [IQR 0.46; 0.78]	0.64 ± 0.30 0.58 [IQR 0.44; 0.83]	0.62 ± 0.28 0.54 [IQR 0.43; 0.74]	0.69 ± 0.35 0.58 [IQR 0.38; 0.79]	0.65 ± 0.30 0.55 [IQR 0.43; 0.75]
-HF	0. 37 ± 0.34	0.19 ±0.35 0.20 [IQR 0.03; 0.38]	0.15 ± 0.38 0.20 [IQR 0.03; 0.38]	0.14 ± 0.39 0.24 [IQR 0.10; 0.44]	0.17 ± 0.37 0.18 [IQR 0.03; 0.35]	0.11 ± 0.35 0.22 [IQR 0.03; 0.36]	0.19 ± 0.34 0.18 [IQR 0.04; 0.35]

* *p* = 0.0000; transfer function results are reported as mean ± SD and median with IQR, interquartile range; VLF: very low frequency; LF: low frequency; HF: high frequency. The reference values of the transfer function estimates are from a previous work of the authors and provided as mean ± SD only; for a more convenient comparison, the transfer function results of the present analysis are therefore provided also as mean ± SD.

**Table 4 diseases-12-00053-t004:** Dynamic cerebral autoregulation parameters in the stroke-unaffected hemisphere.

Variable	Reference Values from [14] in Mean ± SD	No High BP (*n* = 154)	High Blood Pressure (*n* = 183)				
Va		No High BP	Beta-Blockers	Calcium Channel Inhibitors	Diuretics	Renin–Angiotensin–Aldosterone System Inhibitors	
						ACE Inhibitors	AT-1 Receptor Inhibitors
Mean velocity (cm/s)		52 [IQR 44; 65]	45 [IQR 40; 52]	48.5 [IQR 41; 59]	45 [IQR 38; 54]	46.5 [IQR 39.5; 55]	45 [IQR 38.5; 51]
Pulsatility index		0.87 [IQR 0.78; 0.98] *	1.07 [IQR 0.89; 1.26]	1.05 [IQR 0.94; 1.17]	1.05 [IQR 0.89; 1.22]	1.04 [IQR 0.87; 1.13]	1.01 [IQR 0.86; 1.20]
Coherence							
−VLF	0.47 ± 13	56 ± 0.14 0.55 [IQR 0.49; 064]	0.57 ± 12 0.59 [IQR 0.51; 0.72]	0.56 ± 0.11 0.55 [IQR 0.47; 0.67]	0.57 ± 0.13 0.60 [IQR 0.47; 0.71]	0.57 ± 0.14 0.56 [IQR 0.48; 0.68]	0.56 ± 0.11 0.60 [IQR 0.49; 0.70]
−LF	0.73 ± 0.15	0.67 ± 0.15 0.65 [IQR 0.52; 0.78]	0.64 ± 0.15 0.67 [IQR 0.53; 0.74]	0.58 ± 0.14 0.62 [IQR 0.52; 0.75]	0.61 ± 0.15 0.58 [IQR 0.50; 0.74]	0.65 ± 0.13 0.61 [IQR 0.51; 0.74]	0.61 ± 0.15 0.63 [IQR 0.52; 0.76]
−HF	0.65 ± 0.14	0.65 ± 0.14 0.64 [IQR 0.51; 0.75]	0.65 ± 0.14 0.66 [IQR 0.56; 0.76]	0.65 ± 0.16 0.64 [IQR 0.56; 0.75]	0.62 ± 0.16 0.61 [IQR 0.51; 0.73]	0.67 ± 0.15 0.61 [IQR 0.53; 0.75]	0.65 ± 0.15 0.64 [IQR 0.50; 0.76]
Gain (cm/s/mmHg)							
−VLF	0.27 ± 0.29	0.32 ± 0.24 0.26 [IQR 0.16; 0.43]	0.31 ± 0.21 0.27 [IQR 0.14; 0.48]	0.26 ± 0.19 0.26 [IQR 0.20; 0.48]	0.27 ± 0.20 0.31 [IQR 0.13; 0.46]	0.32 ± 0.21 0.28 [IQR 0.18; 0.44]	0.23 ± 0.16 0.27 [IQR 0.15; 0.39]
−LF	0.69 ± 0.41	0.48 ± 0.22 0.47 [IQR 0.30; 0.64]	0.50 ± 0.23 0.42 [IQR 0.32; 0.54]	0.41 ± 0.28 0.44 [IQR 0.31; 0.59]	0.47 ± 0.27 0.43 [IQR 0.34; 0.56]	0.41 ± 0.32 0.37 [IQR 0.30; 0.54]	0.45 ± 0.28 0.44 [IQR 0.35; 0.64]
−HF	0.82 ± 0.51	0.56 ± 0.22 0.55 [IQR 0.42; 0.82]	0.63 ± 0.34 0.52 [IQR 0.38; 0.65]	0.55 ± 0.26 0.52 [IQR 0.42; 0.67]	0.54 ± 0.28 0.52 [IQR 0.36; 0.67]	0.56 ± 0.23 0.55 [IQR 0.40; 0.63]	0.55 ± 0.26 0.52 [IQR 0.40; 0.70]
Phase (radian)							
−VLF	1.12 ± 0.35	0.88 ± 0.43 0.88 [IQR 0.58; 1.20]	0.84 ± 0.39 0.75 [IQR 0.53; 0.99]	0.88 ± 0.38 0.79 [IQR 0.60; 0.99]	0.84 ± 0.39 0.75 [IQR 0.37; 1.05]	0.77 ± 0.36 0.74 [IQR 0.51; 1.00]	0.88 ± 0.35 0.84 [IQR 0.41; 1.25]
−LF	0.74 ± 0.21	0.60 ± 0.25 0.65 [IQR 0.44; 0.87]	0.67 ± 0.38 0.60 [IQR 0.45; 0.80]	0.72 ± 0.32 0.60 [IQR 0.48; 0.79]	0.63 ± 0.27 0.56 [IQR 0.39; 0.72]	0.71 ± 0.32 0.64 [IQR 0.46; 0.95]	0.69 ± 0.32 0.54 [IQR 0.38; 0.73]
−HF	0. 37 ± 0.34	0.19 ± 0.42 0.23 [IQR 0.05; 0.36]	0.22 ± 0.38 0.20 [IQR 0.06; 0.35]	0.16 ± 0.39 0.18 [IQR 0.12; 0.43]	0.20 ± 0.34 0.24 [IQR 0.05; 0.43]	0.14 ± 0.43 0.24 [IQR 0.05; 0.45]	0.21 ± 0.27 0.19 [IQR 0.06; 0.37]

* *p* = 0.0000; transfer function results are reported as mean ± SD and median with IQR, interquartile range; VLF: very low frequency; LF: low frequency; HF: high frequency. The reference values of the transfer function estimates are from a previous work of the authors and provided as mean ± SD only; for a more convenient comparison, the transfer function results of the present analysis are therefore provided also as mean ± SD.

**Table 5 diseases-12-00053-t005:** Univariate regression analysis of predefined factors suggested to predict the acute neurological state on admission as indicated by the National Institute of Health Stroke Scale.

Variable	Adjusted R^2^	Beta	95% CI of Beta	F-Statistic	Univariate Regression *p* =
ACEIs	0.00	0.47	−1.22 2.17	0.29	0.58
BBs	0.03	2.45	0.92 3.97	10.03	0.001
CCBs	0.00	0.23	−1.45 1.92	0.07	0.78
Diuretics	0.02	2.18	0.66 370	8.00	0.005
ARBs	0.00	0.00	−1.52 152	0.00	0.99
Penumbra on CTP (mL)	0.55	0.078	0.07 0.08	359.00	0.0000
Infarct core on CTP (mL)	0.33	0.20	0.17 0.24	145	0.0000
Age	0.02	0.05	−0.01 0.1	6.45	0.01
Systolic BP on admission	0.01	−0.03	−0.0 0.05,	5.23	0.02
Diabetes mellitus	0.009	1.38	−0.12 2.89	3.25	0.07
Estimated glomerular filtration rate (mL/min/1.73m^2^)	0.01	−0.03	−0.002 −0.07	4.44	0.03
Large vessel disease	0.01	1.68	0.10 3.35	3.93	0.04
Coronary artery disease	0.009	1.44	−0.11 3.00	3.32	0.06

ACEI: angiotensin-converting enzyme inhibitor; ARB: angiotensin-1 receptor blocker; BB: beta-blocker; CCB: calcium channel blocker; CTP: computed tomography perfusion; Large vessel disease, ≥50% stenosis or occlusion of the carotid artery system.

**Table 6 diseases-12-00053-t006:** Univariate regression analysis of predefined factors suggested to predict the 3 months outcome as assessed by the modified Rankin scale.

Variable	Adjusted R^2^	Beta	95% CI of Beta	F-Statistic	Univariate Regression *p* =
ACEIs	0.007	0.36	−010 0.83	2.34	0.12
BBs	0.034	0.72	0.30 1.13	11.81	0.0007
CCBs	0.00	0.07	0.39 0.53	0.07	0.75
Diuretics	0.04	0.75	0.34 1.16	13.02	0.0004
ARBs	0.003	0.21	−0.20 0.62	0.98	0.32
Age	0.05	0.022	0.01 0.03	19.29	0.0000
CTP penumbra	0.199	0.013	0.009 0.016	70.82	0.0000
CTP infarct core	0.11	0.032	0.02 0.04	36.20	0.0000
MRI infarct volume	0.23	0.027	0.02 0.03	95.08	0.0000
SysBP on admission	0.001	−0.002	−0.048 0.009	0.36	0.54
IV lysis	0.04	0.70	0.34 1.06	14.77	0.0001
Mechanical thrombectomy	0.14	0.99	0.73 1.25	56.05	0.0000
High blood pressure	0.005	0.23	−0.11 0.58	1.80	0.18
Diabetes mellitus	0.003	0.211	−0.20 0.62	0.98	0.32
eGFR	0.03	−0.01	−0.02 −0.06	10.44	0.001
LVEF	0.015	−0.01	−0.03 −0.002	4.85	0.02
Atrial fibrillation	0.016	0.54	0.08 1.00	5.49	0.01
NIHSS on admission	0.40	0.17	0.14 0.19	219.00	0.0000
Large vessel disease	0.001	0.263	−0.19 0.71	0.49	0.48
AHPhaseVLF	0.003	−0.37	−0.82 0.08	2.60	0.10
AHPhaseLF	0.002	−0.23	−0.77 0.31	0.70	0.40

ACEI: angiotensin-converting enzyme inhibitor; AHPhaseVLF/LF, phase (radian) in the very low/low frequency range in the stroke-affected hemisphere; ARB: angiotensin-1 receptor blocker; BB: beta-blocker; CCB: calcium channel blocker; CTP: computed tomography perfusion, penumbra and infarct core per mL; eGFR, estimated glomerular filtration rate (mL/min/1.73 m^2^); IV lysis, intravenous thrombolysis; Large vessel disease: ≥50% stenosis or occlusion of the carotid artery system; LVEF:, left ventricular ejection fraction (%); MRI: magnetic resonance imaging, infarct volume per mL; SysBP, systolic blood pressure, per mmHg; NIHSS, National Institute of Health Stroke Scale.

## Data Availability

All data is available on request from the corresponding author.

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
