# Peer review of "Pre-Stroke Antihypertensive Therapy Affects Stroke Severity and 3-Month Outcome of Ischemic MCA-Territory Stroke"

_diseases, 2024, doi:10.3390/diseases12030053_

Round 1
Reviewer 1 Report
Comments and Suggestions for Authors Dear Editor of Diseases – MDPI Thank you for inviting me to be a Referee for this scientific article: “Pre-stroke antihypertensive therapy affects stroke severity and 3 months outcome of ischemic MCA-territory stroke”. - Despite my opinion transcription regarding what is requested in the electronic form Diseases – MDPI, I would like to add some comments. - There are also some aspects that deserve changes that are referenced in red: (in red purposals to be potentially changed and also to be praised). Abstract: - Lines 10-11: it says: “Among 337 consecutive ischemic stroke patients (female 102; median age 71years [25%, 75% percentile 60, 78]; NIHSS median 3 [1.6]) with assessment of dCA … ”; Is this the best way to present the results? When it is written “percentile 60, 78]; NIHSS median 3 [1.6])”; It may look like quotes [60,78] and quotes [1,6] wouldn't it be better to use curved parentheses ( )? - Lines 18-19: it says: “the likelihood of a less severe stroke (Odds ratio 0.691, 95%CI 0.493 0.972; p=????), and that beta-blockers decreased the likelihood of a better modified Rankin score at 3 months (0.981, 0-970 0.992; p=???”. - Lines 19-21: “it says: “. Other independent factors associated with stroke severity were age, penumbra and infarct volume. Other independent factors associated with stroke outcome were penumbra and infarct volume”; repeated speech - delete one of the sentences.Good and adequate introduction. - -Line 48: “patients with HBP. ]; without ] - - Lines 52-53:” Dynamic cerebral autoregulation …CA assessment is a routine procedure in our stroke unit…”; an undeniable added value of this study. - -Lines 54-56: Objectives…it is written “Aim of the study, we sought to determine whether different classes of antihypertensive drugs impact dCA differently and whether this corresponds to a reduced stroke severity and better three-month outcomes.”; excellent and original objectives, given the scientific uncertainty/ignorance in this very important area of human pathology. Methods Lines 58-129: Generally well-described and appropriate methodology as well as patient inclusion/exclusion criteria. However: - - the methodology to evaluate “dynamic cerebral autoregulation” should be described in more detail and not simply indicate the bibliographic citations [14,15], namely, which method was used to force oscillations in BP…( squat-stands or sit- stand, a random inflation/deflation of thigh cuffs, single thigh cuff deflation head-up tilt, cold pressor test, hand grip, the Valsalva maneuver?), other or none?. - - Lines 92-93: it says “…the presence of a characteristic hemispheric syndrome diagnosed as a supratentorial ischemic stroke in the middle cerebral artery (MCA) territory after initial multimodal”; MCA in which segments…M1, M2?? And if possible, what was the distribution between the different groups of antihypertensives? - Line 96: “…with the stroke, AH)”; with the stroke, (AH)
- -Line 98: it says: “and determination of dCA within 48 hours of the stroke event”; but this determination remains reliable and biologically/medically important depending on the objectives of the study-48 h later??? How does dCA usually evolve in the first hours after Stroke? How many cases were evaluated after the first 24 hours and what influence did this have on dCA evaluations? - Lines 100-103: it says “Regarding the pre-stroke treatment of their BPH, we classified the patients into those having received either diuretics, or BB, or CCB, or ACEI or ARB regardless of whether the patients received additional antihypertensive medication of another class,”; How many were taking 2 or 3 or more antihypertensives? - -Line 108: “continuously and simultandfghously”; must be simultaneously. - Lines 115-120: it says “The analyzed frequency range was 0.02–0.50 Hz; the results are reported as their respective averages in the very low frequency range (VLF, 0.02-0.07 Hz), low frequency range (LF, 0.07-0.15 Hz), and high frequency range (HF, 0.16-0.5 Hz)”; however, other values are referenced in the literature – Panerai et al (2023)“Presenting the complete frequency dependence of coherence, gain and phase in the range 0.02–0.5 Hz, as mean and standard deviation values, is important when reporting TFA results. Until further evidence is available, statistical analyzes should be based on averaged values within the very low-frequency (0.02–0.07 Hz), low-frequency (0.07–0.2 Hz) and high-frequency (0.2–0.5 Hz) bands” In: Transfer function analysis of dynamic cerebral autoregulation: A CARNet white paper 2022 update Ronney B Panerai, Patrice Brassard, Joel S Burma, et al; on behalf of the Cerebrovascular Research Network (CARNet); Journal of Cerebral Blood Flow & Metabolism 2023, Vol. 43(1) 3–25 Results - Lines 131-132: it says: “During the referred time span 337 patients (102 female; median age 71 years [25%, 75% percentile IQR?N60, 78]…what does this mean???; NIHSS median 3 [1,6])…here it seems that there are quotes from the authors of article 1 and article 6 – this aspect should be modified… ; I don't know if, for example, [IQR?,1 6] in the text referring to Interquartiles is not to be confused with bibliographic citations with the same numbers. - -Lines 133-134: it says “220 patients had a minor (NIHSS ≤5), 82 a mild to moderate (NIHSS 6-15), and 35 a major (NIHSS ≥16) stroke [16]; but in the cited article by Koton et al it is mentioned “Stroke severity was classified into 4 levels: minor (NIHSS ≤5), mild (NIHSS 6-10), moderate (NIHSS 11-15), and severe (NIHSS ≥16)” ; Why this discrepancy in the categorization of stroke severity? The content of the tables must be reviewed. Therefore, it would be important to add a line above No High Blood Pressure and above the different classes of antihypertensives with the phrase: High Blood Pressure patients, n=183; Perhaps there are no columns to say how many patients took 2 antihypertensives or 3 or more… Were beta blockers and calcium antagonists only used as antihypertensives or also as antianginals? and ACEi and ARB were only used as antihypertensives or in the post-myocardial infarction period of patients who had a stroke?. For all this, it is important to emphasize the importance of placing the High Blood Pressure patients, n=183 line above the different classes of drugs. - - In Table 1: “Baseline characteristics of the study population of stroke patients without and with high blood pressure (BP). General medical aspects”. - Wrong title of the table – there is no data on hypertensive patients (and it is very important to understand that they are the ones taking the different classes of antihypertensives) (see the proposal above). - - Line Age /No Hihg BP: 62.5 [51 72]*: mean and standard deviation or interquatile??? Wouldn't 62.5 (51-72)* be better and so on for all values in the tables? What it means *? - not in the caption - N= ?? must be N only - Line Diabetes mellitus: No Hihg BP 23**…but then column Beta-Blocker N=27 (the data presentation is very confusing???) - Coronary artery disease?: what does it mean-definition?… - Left ventricular ejection fraction… Left (space) ventricular ejection fraction o Antihypertensive therapy with 0/1/2/3(>3?? Confusing presentation o Why are these data in the Table?? What's the point????Kruskal-Wallis Test: df 5, chi2 64.02, 153 p=.0000; **, Chi2 test: df 5, chi2 14.03, p=.01; isn't it enough to put p< 0.000?; because the Kruskal-Wallis Test is referenced in statistical methods… o Line Table 1 “Antihypertensive therapy with 0/1/2/3/>3”; What does it mean for example: 0/17/29/22/8 …in the 3rd column from the left …and in the other columns? - Lines 152-154: it says “ACE, Angiotensin Converting Enzyme”; should be ACE: Angiotensin Converting Enzyme as well as the other acronym subtitles and : - Table 2. “Baseline characteristics of the study population of stroke patients without and with high blood pressure (BP). Neurological aspects Various inaccuracies/omissions: - What does “Frequencies of transient ischemic attack/stroke” mean?; frequencies or N 21/134 or 21/133 in the 4th line2nd column - Wouldn’t the presentation of the National Institute of Health Score (NIHSS) at entry be better expressed as median (IQR)?; for example, or the value and IC? In parentheses…In the High BP column: instead of 3 [1 3.5]…be 3 (1 – 6.5) and so on for all remaining columns? The same can be said for expressing the results on Modified Rankin scores (mRs) at entry? - The line about CTP infarct core (ml) is also not well understood and not well expressed: “ 0[0 0] ???range??; the same for CTP penumbra (ml)??, MRI infarct size…p values??? - Of patients that did Intravenous lysis (Total = 123): 56 were non-hypertensive and the rest were hypertensive???; the same reasoning for those who underwent mechanical thrombectomy…review the way of expressing the results with regard to the other lines in this table 2. - Lines 160-163: it says: “Regarding dCA, there were no differences in phase or gain between the different groups with or without HBP (Supplemental Tables 1 and 2). To indicate the best dCA range, the upper quartile can be used. For phase in the VLF /LF, the upper quartile is above 1.01/0.78 radians in both hemispheres… … The corresponding gain values are < 0.44/0.57” ; but the main objective of this study, as defined in Lines 54-56 is: “Aim of the study, we sought to determine whether different classes of antihypertensive drugs impact dCA differently and whether this corresponds to a reduced stroke severity and better three-month outcomes.” , so these results must be clearly expressed and not in supplementary tables…. (these tables must be present in the main text as dCA belongs to one of the main objectives of this study). It is also suggested, to enrich this work: - Normal reference values for each parameter must be included in these “supplementary” tables (see, for example, Tables 3 and 4, published in: [15] :Transfer function analysis of dynamic cerebral autoregulation: A CARNet white paper 2022 update Ronney B Panerai, Patrice Brassard, Joel S Burma, et al., on behalf of the Cerebrovascular Research Network (CARNet); Journal of Cerebral Blood Flow & Metabolism 2023, Vol. 43(1) 3–25; - Place the phrase above the different classes of antihypertensives: High Blood Pressure patients, n=183 (even in the “supplementary “tables) - The authors have data on ARI (autoregulatory index) values? Was there any reason not to do it? -Present the different values related to dCA and calculate statistical data among all hypertensive and non-hypertensive patients (without classifying them by classes of antihypertensives such as medication) (and also extend this approach – hypertensive and non-hypertensive….to the NIHSS and mRankin values scale. - Correct caption is missing (VLF; LF…) and remember that these are parameters related to brain autoregulation flow in order to better understand the content of Lines 193-196). - Lines 175-185: it says “In a multinomial ordinal logistic regression analysis model with the stroke categories minor, moderate and major stroke the dependent variable and age, BB, diuretics, large vessel disease, penumbra and stroke core volume on CTP, Systolic BP on admission, and eGFR the independent variables, the odds ratios to be allocated in the better stroke category (minor vs mild to moderate vs major) were: 0.978 [0.966 0.992] per each increasing year of age, 0.970 [0.960 0.974] per increasing ml penumbra on CTP, 0.973 [0.957 0.981] per increasing ml stroke volume core on CTP, 0.997 181 [0.991 1.00] per increasing mmHg systolic BP on admission, 0.992 [0.981 1.052] per increasing ml eGFR, 1.45 [0.951 2.021] for the presence of extracranial large vessel disease, 0.845 [0.592 1.087] for the pre-stroke use of Beta-Blockers, and 0.691 [0.493 0.972] for the pre-stroke use of diuretics.” Regarding this text (lines 175-185), it is suggested to change: Systolic to systolic; (minor vs mild to moderate vs major) by (minor vs mild to moderate vs major, according to NIHSS); in this result only one value appears and compares 3 situations: minor vs mild to moderate vs major…???; odds ratios – put (OR 95% CI) to be allocated in the better stroke category (minor vs mild to moderate vs major) were: 0.978 [0.966 0.992]- the p value? For this variable and all others; discuss further the results found depending on the p value. Probably removing the results referring to : 1.45 [0.951 2.021], p=?for presence of extracranial large vessel disease, 0.845[0.592 1.087], p=? for the pre-stroke use of Beta-Blockers, and 0.691 [0.493 0.972], p=? for the pre-stroke use of diuretics.”,p=?, will the results of the other variables be significant? Comment later in the discussion chapter…. - It would be important to present compared results of non-hypertensive patients with hypertensive patients (including 183 patients): NIHSS, mRankin sclale and dCA (with statistical treatment). - Lines189-192: caption…place acronyms in alphabetical order and acronym, for example ACEI, Angiotensin Converting Enzyme inhibitor… ACEI: Angiotensin Converting Enzyme inhibitor…in fact, as should be changed in all captions of all tables…instead of acronym and , must be an acronym and: - Lines 192 and 193: separate the lines - Lines 207-214: it says “the odd-ratios to be allocated in the better mRs group were: 0.981 [0-970 0.992] for the use of BB, 0.721 [0.517 1.004] for the use of diuretics, 1.10 [0.785 1.562] for each year of age, 1.005 [1.002 1.008] for each ml penumbra, 0.984 [0.978 0.998] for each ml stoke volume on CTP, 0.982 [0.976 0.988] for each ml MRI stroke volume, 1.24 [0.930 1.672] for the use of i.v. 0.755 0.801] per each point on the NIHSS scale. Thus, over the entire mRs range most independent variables showed an odds ratio to be in the poorer mRs group.” Regarding this text (lines 207-214), it is suggested: odds ratios – enter (OR 95% CI), and p value following the confidence intervals; then discuss in the respective chapter what was associated with worse and better outcomes. - Lines 218-224: it is written: “We explain the penumbra finding as a result of an aggressive lysis strategy (iv and mechanical) from which patients can benefit. Patients who undergo thrombectomy are clinically in a severe neurological state which will lead to a poor mRs group if the procedure is not performed; because recovery from the severe deficit is most often not complete, with still a poorer outcome compared to patients with a minor stroke, we see the thrombectomy result in our analysis as a marker of the baseline stroke severity which then turns out in a poorer mRs group .”; this text should be transferred to the discussion (and not be in the results chapter). The discussion should also be extended to the results of other authors on thrombectomy in the treatment of Stroke (such as: Masthof, M, KrählingH, Han-Akkurt B, et al. Evaluation of effectiveness and safety of the multizone NeVaTM stent retriever for mechanical thrombectomy in ischemic stroke.Neuroradiology (2023) 65:1777–1785). - In this original work it is not mentioned whether there were cases of hemorrhagic transformation (Intracranial Hemorrhage at 24 Hours - as in the work by Maier et al) and whether this affected the outcomes; if affected, which group of antihypertensives was most associated with a worse outcome (see Maier et al 2022 who cite as a reference [18]…who conclude that “we showed an opposite effect of baseline AHT, based on their effect on the RAS. Patients treated with RAS inhibitor agents before AIS exhibited less severe AIS compared with patients under non-RAS inhibitor treatments, developed less intracranial hemorrhage at 24 hours and had a trend toward better NIHSS score at 24 hours”; note – Maier et al (2022) ” We categorized these treatments as RAS inhibitors (ACE inhibitor, ARB, and BB) and non-RAS inhibitors (CCB and diuretics), as previously published.” Discussion The discussion is too succinct and could be improved. - Line 239: it says “The antihypertensive pre-stroke medication does not affect dCA.” This result should be discussed in comparison with what is well described in the literature, for each class of antihypertensive, such as that published by Llwyd et al (2022) that the authors cite as a reference [4]. - The discussion should be expanded because the dCA measurement was carried out up to 48 hours post-admission and there were no changes…contrary to what is described, for example, by several authors, such as what can be found in the article by Nogueira et al cited as [17]; The hypothesis that the suspension of antihypertensive drugs upon admission had affected the measurement of dCA was also not discussed (5 plasma half-lives have passed – duration of action of the drugs?). - Line 243: it says “hemodynamic difference was that the pulsatility index of the CBFV in the hypertensive”; but in the article cited as [14] there is an analysis of cerebrovascular resistance and not pulsatile index (probably related); How do HBP and age affect this parameter? - Lines 246-252: it says “In our patients, the antihypertensive pre-stroke medication with diuretics did influence stroke severity in the logistic regression model.”; influenced in what sense? But beta blockers also negatively influenced the NIHSS (Stroke severity); but the fact that there was this negative association (statistically significant), was it that significant and with potential clinical impact? These aspects remain to be discussed - Lines 253-255: it says “Maier et al [18] reported that the patients using diuretics and CCB had a more severe NIHSS at admission than the non-user of diuretics [Maier et al: RAS inhibitors (ACE inhibitor, ARB, and BB) and non-RAS inhibitors (CCB and diuretics)]; in this?? (Maier et al or in the current study? pathophysiological explanation in the article by Maier et al [18]…. - Lines 264-265: it says “Amongst the antihypertensive pre-stroke medication classes, the use of BB was associated with a poorer outcome compared to CCB, diuretics, ACEI, and ARB.”; which authors claim this or demonstrate this?; in the current study “the odd-ratios to be allocated in the better mRs group were: 0.981 [0-970 0.992] for the use of BB, 0.721 [0.517 1.004] for the use of diuretics” (p=0.0004)…this aspect of negative association with the outcome is not discussed or when this discussion occurs, it is not done in the most correct way (“Amongst the antihypertensive pre-stroke medication classes, the use of BB was associated with a poorer outcome compared to CCB, diuretics, ACEI, and ARB.”); but the use of diuretics is also associated with worse outcome in the present study; - Lines 266-268: it says “Meier et al [18] reported protective effects of ACEI and CCB in patients undergoing mechanical thrombectomy”; endovascular therapy - no results in Maier et al study.???”; instead of Meier it should be Maïer et al. - In the present study it can be read (line 211) “0.542 [0.439 0.671] for the use of mechanical thrombectomy”; any other explanation other than the one mentioned (for example with the technique used?; compare with other results from other publications) - Lines 295-301: it says “Arterial hypertension is a major stroke risk factor, and its therapy is an essential stroke prevention strategy. Whether the one or the other antihypertensive drug class will provide a lower risk of a poorer stroke outcome is under debate. In our cohort of stroke patients those who had a diuretic in the pre-stroke HBP management fared worse in the acute stroke condition, and those who had a BB in its pre-stroke medication exhibited a poorer mRs score three months after the stroke event. ”; but not only patients taking diuretics but also BB exhibited worse in the acute stroke condition…; “those who had a BB in its pre-stroke medication exhibited a poorer mRs score three months after the stroke event” or those who used diuretics instead BB? References - It is: 1. 1.Rodríguez-Yañez M, Gómez-Choco M, López-Cancio E, et ad hoc committee of the Spanish Society of Neurology's Study Group for Cerebrovascular Diseases. Stroke prevention in patients with arterial hypertension: Recommendations of the Spanish Society of Neurology's Stroke Study Group. Neurology (Engl Ed). 2021 Jul-Aug;36(6):462-471. doi: 10.1016/j.nrleng.2020.04.023. 311 Epub 2021 Apr 23. PMID: 34238528. - It must be: 1. Rodríguez-Yañez, M.; Gómez-Choco, M.; López-Cancio, E.; et al. Stroke prevention in patients with arterial hypertension: Recommendations of the Spanish Society of Neurology's Stroke Study Group. Neurology (Engl Ed). 2021 Jul-Aug;36(6):462-471. doi: 10.1016/j.nrleng.2020.04.023. 311 Epub 2021 Apr 23. PMID: 34238528. References must all be reviewed…in accordance with the above. - The number is repeated…. 1. and 1, 2 and 2 and so on. - It is: 5 Tziomalos K, Giampatzis V, Bouziana SD, et. Effects of different classes of antihypertensive agents on the outcome of acute ischemic stroke. J Clin Hypertens (Greenwich). 2015 Apr;17(4):275-80. doi: 10.1111/jch.12498. Epub 2015 Mar 13. PMID: 323 25765927; PMCID: PMC8031997.”; There is a lack of et al and why not 10 authors and then et al when appropriate? (review all references regarding these aspects as well) In conclusion: This article under analysis is of scientific interest, but some of the aspects need to be reformulated (see criticism) and the discussion should be expanded and improved.

Reviewer 2 Report
Comments and Suggestions for Authors
I read with interest the manuscript of Lakatos et al. They evaluated whether different antihypertensive drug classes in high blood pressure (HBP) pre-stroke treatment affect dynamic cerebral autoregulation (dCA), stroke severity, and outcome among 337 consecutive ischemic stroke patients. They concluded that the pre-stroke antihypertensive treatment with diuretics was associated with a more severe neurological deficit on admission and pre-stroke treatment with beta-blockers with a poorer 3-month outcome.
I think the manuscript will be of high interest to the readership and will convey an important message. However, I believe the results should be illustrated by at least one figure. For example, the authors may add a plot illustrating the regression analysis or plot with the differences of events in patients with/without diuretics and beta-blockers.
Reviewer 3 Report
Comments and Suggestions for Authors
The authors evaluated 337 patients with ischemic stroke patients (female 102; median age 71 years [25%, 75% percentile 60, 78]; NIHSS median 3 [1,6]) and assessed the impact of various antihypertensive drugs used in the period before the onset of the disease on their impact on the severity of the disease. stroke and results after 3 months.
The authors found that pre-stroke diuretic use influenced stroke severity in a logistic regression model, and in univariate regression analysis, beta-blocker (BB) use predicted a worse NIHSS score at admission.
In their conclusions, the authors concluded that the use of BB and/or diuretics in the group of analyzed patients with mild to moderate stroke had a small effect on the severity of stroke of BB and diuretics compared to the importance of penumbra and infarct size in initial cranial perfusion. Additionally, the authors did not find a positive effect of angiotensin-converting-enzyme inhibitors (ACEI), or angiotensin-1 receptor blockers (ARB), which differs from the results in animal models, which showed a positive protective effect.
In a further conclusion, the authors concluded that although the use of calcium channel blockers (CCB) and renin-angiotensin system inhibitors may have protective effects on cerebral blood vessels in other studies, stroke invalidates any measurable effect of drugs on dynamic cerebral autoregulation.
The results of this study show that despite the positive effect of ACE and BB drugs, e.g. in patients with myocardial infarction, there may be a change in the preferences of the drugs used in the event of brain injury of ischemic etiology, which is important information resulting from the above work.
Round 2
Reviewer 2 Report
Comments and Suggestions for Authors
The authors addressed all raised comments and the manuscript has improved.